# Phase 2 Trial (POLA Study) of Lurbinectedin plus Olaparib in Patients with Advanced Solid Tumors: Results of Efficacy, Tolerability, and the Translational Study

**DOI:** 10.3390/cancers14040915

**Published:** 2022-02-12

**Authors:** Andres Poveda, Raquel Lopez-Reig, Ana Oaknin, Andres Redondo, Maria Jesus Rubio, Eva Guerra, Lorena Fariñas-Madrid, Alejandro Gallego, Victor Rodriguez-Freixinos, Antonio Fernandez-Serra, Oscar Juan, Ignacio Romero, Jose A. Lopez-Guerrero

**Affiliations:** 1Oncogynecologic Department, Initia Oncology, Hospital Quironsalud, Avda Blasco Ibañez, 14, 46010 Valencia, Spain; 2Laboratory of Molecular Biology, Fundación Instituto Valenciano de Oncología, 46009 Valencia, Spain; rlopez@fivo.org (R.L.-R.); afernandez@fivo.org (A.F.-S.); jalopez@fivo.org (J.A.L.-G.); 3IVO-CIPF Joint Research Unit of Cancer, Príncipe Felipe Research Center (CIPF), 46012 Valencia, Spain; 4Medical Oncology Department, Vall d’Hebron University Hospital, Vall d´Hebron Institute of Oncology (VHIO), 08035 Barcelona, Spain; aoaknin@vhio.net (A.O.); lfarinas@vhio.net (L.F.-M.); victor.rodriguezfreixinos@sunnybrook.ca (V.R.-F.); 5Medical Oncology Department, Hospital Universitario La Paz-IdiPAZ, Universidad Autónoma de Madrid (UAM), 28049 Madrid, Spain; andres.redondos@uam.es (A.R.); alejandro.gallego@salud.madrid.org (A.G.); 6Medical Oncology Department, Universitary Hospital Reina Sofia, 14004 Cordoba, Spain; mjesusrubio63@gmail.com; 7Medical Oncology, Hospital Universitario Ramón y Cajal, 28034 Madrid, Spain; eva_m_guerra@hotmail.com; 8Department of Medical Oncology and Hematology, Odette Cancer Centre, Sunnybrook Health Sciences Centre, Toronto, ON M4N 3M5, Canada; 9Medical Oncology, Pivotal SLU, 28023 Madrid, Spain; oscar.juan@pivotalcr.com; 10Medical Oncology, Fundacion Instituto Valenciano de Oncologia, 46009 Valencia, Spain; iromero@fivo.org; 11Department of Pathology, School of Medicine, Catholic University of Valencia ‘San Vicente Mártir’, 46001 Valencia, Spain

**Keywords:** ovarian cancer, endometrial cancer, lurbinectedin, olaparib, genomic instability

## Abstract

**Simple Summary:**

Genomic instability (GI) is a transversal phenomenon in oncology, constituting a hallmark of cancer. In gynecological malignancies, the predictive value of GI has been described and is mainly caused by alterations in the homologous recombination repair (HRR) genes, such as *BRCA1/2*. The POLA clinical trial constitutes an ideal substrate used to study the correlation between GI and response to combined therapy of lurbinectedin plus olaparib in solid tumors. In this context, we developed an approach based on next-generation sequencing, capable of shedding information about Copy Number Variations (CNV) as a surrogate of GI and genotyping of homologous recombination repair genes. Additionally, some algorithms used to extract GI parameters were tested and benchmarked, selecting the most informative mutational and GI features as potential predictive biomarkers for the drug combination explored in the POLA trial.

**Abstract:**

We hypothesized that the combination of olaparib and lurbinectedin maximizes DNA damage, thus increasing its efficacy. The POLA phase 1 trial established the recommended phase 2 dose of lurbinectedin as being 1.5 mg (day 1) and that of olaparib as being 250 mg/12 h (days 1–5) for a 21-day cycle. In phase 2, we explore the efficacy of the combination in terms of clinical response and its correlation with mutations in the HRR genes and the genomic instability (GI) parameters. Results: A total of 73 patients with high-grade ovarian (*n* = 46), endometrial (*n* = 26), and triple-negative breast cancer (*n* = 1) were treated with lurbinectedin and olaparib. Most patients (62%) received ≥3 lines of prior therapy. The overall response rate (ORR) and disease control rate (DCR) were 9.6% and 72.6%, respectively. The median progression-free survival (PFS) was 4.54 months (95% CI 3.0–5.2). Twelve (16.4%) patients were considered long-term responders (LTR), with a median PFS of 13.3 months. No clinical benefit was observed for cases with HRR gene mutation. In ovarian LTRs, although a direct association with GI and a total loss of heterozygosity (LOH) events was observed, the association did not reach statistical significance (*p* = 0.055). Globally, the total number of LOHs might be associated with the ORR (*p* =0.074). The most common grade 3–4 toxicities were anemia and thrombocytopenia, in 6 (8.2%) and 3 (4.1%) patients, respectively. Conclusion: The POLA study provides evidence that the administration of lurbinectedin and olaparib is feasible and tolerable, with a DCR of 72.6%. Different GI parameters showed associations with better responses.

## 1. Introduction

The treatment of ovarian cancer has seen increasing improvement over recent years. Today, the most critical advance has been the use of poly (ADP-Ribose) polymerase inhibitors (PARPis). In 2009, a phase I study on olaparib presented the first clinical evidence of PARPi having an effect in patients with *BRCA1/2* mutations, with the benefits being of a magnitude never observed before [1]. However, the clinical benefit of PARPi is not limited to patients with *BRCA1/2* mutations; the entire population of high-grade serous (HGS) ovarian cancer or triple-negative breast cancer has observed its benefits [2]. In patients with the absence of *BRCA* alterations, the efficacy of PARPi is more pronounced in those with homologous recombinant deficiency (HRD). Several phase II and III trials have demonstrated the efficacy of PARPi in patients with ovarian cancer [3] and have led to the approval of three PARPis—olaparib, niraparib, and rucaparib—as maintenance therapy for platinum-sensitive recurrent ovarian cancer [4,5,6,7,8].

Regarding maintenance, PARPis have been administered as monotherapy in a continuous oral dosing schedule. The combination of PARPis with chemotherapy or other agents is an interesting approach to increasing their efficacy, especially in previously treated patients and those with unknown DNA repair deficits [9]. However, combination trials using continuous olaparib with chemotherapy, such as cisplatin or carboplatin alone, or combined with gemcitabine, had to be stopped prematurely due to high hematological toxicity. An intermittent dose of olaparib, especially with a short course, has shown better tolerability when combined with chemotherapy [10,11,12,13]. Myelosuppression is the main effect caused by toxicity of PARPi and is considered a “class toxicity”. However, only the PARPi veliparib has been associated with lower myelosuppression in monotherapy studies, and the continuous administration of veliparib has been successfully combined with chemotherapy [14,15].

Trabectedin is an anticancer drug structurally related to ecteinascidins and approved in many countries to treat patients with relapsed platinum-sensitive ovarian cancer. Lurbinectedin (PM01183) is a novel synthetic alkaloid structurally similar to trabectedin. Both contain a pentacyclic skeleton composed of two fused tetrahydroisoquinoline rings (subunits A and B) responsible primarily for DNA recognition and binding. However, the additional module (ring C) in lurbinectedin is a tetrahydro β-carboline rather than the additional tetrahydroisoquinoline present in trabectedin. This structural difference may confer pharmacokinetic benefits and intrinsic activity [16]. Lurbinectedin joins covalently to the DNA, inducing DNA double-strand breaks that initiate apoptosis [17] and reducing tumor-associated macrophages and the inflammatory microenvironment by inhibiting inflammatory factors [17]. Lurbinectedin has been recently approved by the U.S. Food and Drug Administration (FDA) based on a phase 2 single-arm study in 105 platinum-sensitive and platinum-resistant adult patients with metastatic small cell lung cancer and disease progression on or after platinum-based chemotherapy [18]. In a small randomized phase 2 study, lurbinectedin showed high activity in patients with platinum-resistant ovarian cancer [19]. However, a recently published phase 3 study contradicted these results, with lurbinectedin showing a similar antitumor efficacy to topotecan or liposomal doxorubicin but having a better toxicity profile [20].

Combining a PARPi (olaparib) with a DNA damaging agent (trabectedin or lurbinectedin) is an exciting approach to maximizing the effect of DNA damage. In preclinical models, the combination of both agents was synergistic and led to biologically significant deregulation of the DNA damage repair machinery that elicited relevant antitumor activity [21,22]. However, overlapping hematological toxicity may represent a limitation of the combination. The lurbinectedin dose adjusted to the body surface area showed lower hematological toxicity (57%) than flat dose [19,23]. Similarly, an intermittent schedule of olaparib is feasible and has a lower rate of hematologic adverse events than a continuous course when combined with chemotherapy [10,11].

Recently, we reported the results of a phase I dose-finding study with a short course of olaparib and lurbinectedin in patients with ovarian and endometrial cancer. The dose-limiting toxicity was grade 4 neutropenia, and the recommended phase 2 dose (RP2D) was 1.5 mg/m² of lurbinectedin administered intravenously on day 1 and 250 mg of olaparib administered as oral capsules twice a day (BID) on days 1–5 of a 21-day cycle [24]. Most adverse events were mild, and the treatment was well-tolerated. Moreover, we obtained a disease control rate (DCR) of 60% (but no responses). Overall, the favorable safety profile and preliminary efficacy results deserved further investigation.

The POLA is the first phase 2 trial to assess the efficacy and toxicity of lurbinectedin and olaparib in previously treated gynecological tumors and their correlation with molecular characteristics.

## 2. Materials and Methods

### 2.1. Study Population

This is a phase 2, open-label, non-randomized study that recruited patients from five centers in Spain. Patients aged ≥ 18 years were eligible if they had histologically confirmed advanced or metastatic HGS or endometroid (no mucinous and no clear cells) platinum-resistant—Not refractory (neither primary nor secondary)—Ovarian cancer, fallopian cancer, primary peritoneal cancer, endometrial cancer (any grade, not platinum-refractory), or triple-negative breast cancer; had an Eastern Cooperative Group (ECOG) performance status (PS) ≤ 2; had a life expectancy of ≥3 months; had a measurable disease according to the Response Evaluation Criteria in Solid Tumors (RECIST) version 1.1; received at least one line of standard therapy for locally advanced or metastatic disease and developed progression disease afterwards (no limit was placed on the number of prior therapies); had hemoglobin ≥ 10 g/dL; had an absolute neutrophil count ≥ 1500/µL; had platelets ≥ 100,000/µ; had total bilirubin ≤ 1.5 times the institutional upper limit of normal (ULN); had aspartate aminotransferase and alanine aminotransferase ≤ 2.5 times ULN; had albumin ≥ 3 g/dL; and had creatinine ≤ 1.5 times the ULN or a creatinine clearance ≥ 30 mL/min. Patients were ineligible if they had received previous treatment with a PARPi or lurbinectedin.

The study (NCT02684318, EudraCT 2015-001141-08, 03.10.2015) was approved by a centralized ethics committee and was conducted following the Declaration of Helsinki, ICH Good Clinical Practice guidelines, and the current legislation. Written informed consent was obtained from all patients before conducting study-specific procedures.

### 2.2. Study Treatment

The patients received 1.5 mg/m^2^ of lurbinectedin intravenously on day 1 in combination with oral administration of 250 mg of olaparib/12 h on days 1–5 BID of a cycle of 21 days according to the RP2D determined in the phase I trial. The study treatments were given until objective disease progression according to the RECIST 1.1, unacceptable toxicity, or patient withdrawal of consent. At screening, patients underwent a history and physical examination, baseline hematological and chemistry assessments and urinalysis, blood sampling for pharmacogenomics (PG) analysis, ECG, and tumor assessment. The patients were seen on day 1 and day 15 of cycles 1 and 2 and every 3 weeks for the rest of the cycles for history and physical examination, hematological and chemistry assessment, and PG sampling (only in cycle 1). Tumor response was assessed by the investigators using the same method used during screening, which was in line with the RECIST v1.1 every 2 cycles (6 weeks) until disease progression or death. All toxic effects were graded using the National Cancer Institute-Common Terminology Criteria for Adverse Events (NCI-CTCAE) version 4.0.3 (https://ctep.cancer.gov/protocoldevelopment/electronic_applications/ctc.htm#ctc_40 (accessed on 22 January 2022)).

### 2.3. Outcomes

The primary endpoint was overall response rate (ORR), defined as a complete response (CR) or partial response (PR) according to the RECIST v1.1. The secondary endpoints were progression-free survival (PFS), overall survival (OS), safety, and translational studies. The exploratory objectives included ORR and PFS by tumor type and by the number of previous treatment lines, duration of response, and long-term responders (LTR).

### 2.4. Translational Studies

#### 2.4.1. DNA Extraction

DNA extraction was performed using 3 × 20 μm sections of formalin-fixed paraffin-embedded (FFPE) archived tumors and the QIAmp DNA FFPE Tissue kit (Qiagen Iberica S.L., Spain). DNA integrity, concentration, and fragment size were determined using a Genomic DNA ScreenTape assay (TapeStation 4200, Agilent, Santa Clara, CA, USA).

#### 2.4.2. Next-Generation Sequencing (NGS) Panel

The libraries were prepared using the Agilent (Santa Clara, CA, USA) SureSelect-XT HS Target Enrichment Kit combined with OneSeq backbone 1 Mb. Briefly, 200 ng of extracted DNA were enzymatically fragmented to a size between 150 and 200 bp. Each library was then hybridized to a SureSelect custom panel (Agilent) according to the manufacturer’s protocol. The custom panel, designed to evaluate the HRD status, includes 35 genes involved in different DNA repair pathways: *BRCA1*, *BRCA2*, *BARD1*, *BRIP1*, *CHEK1*, *CHEK2*, *FAM175A*, *NBN*, *PALB2*, *ATM*, *MRE11A*, *RAD51B*, *RAD51C*, *RAD51D*, *RAD54L*, *FANCI*, *FANCM*, *FANCA*, *ERCC1*, *ERCC2*, *ERCC6*, *REQL*, *XRCC4*, *HELQ*, *SLX4*, *WRN*, *ATR*, *PTEN*, *CCNE1*, *EMSY*, *TP53*, *MLH1*, *MSH2*, *MSH6*, and *PMS2*, and 147,000 SNPs distributed homogenously along the genome that served to obtain Copy Number (CN) profiles. The pooled library was sequenced (2 × 100 cycles) on a NextSeq550 using a high output flow cell (Illumina, San Diego, CA, USA). A secondary analysis was performed with Haplotypecaller (Broad Institute) for variant calling and Variant studio 4.0 for annotation (Illumina, San Diego, CA, USA). The variants were considered when classified as pathogenic (P), likely pathogenic (LP), or variant of unknown significance (VUS) with pathogenic prediction or variants with both in silico predictors, SIFT and Polyphen, predicted as pathogenic. The variants were filtered based on the coverage and functional annotation. The minimum coverage for a variant was established at 100×. Mutations were accepted with a frequency higher than 5%. For CN calling at the gene level, the PanelMops package [25] from R was applied. The genomic instability was established using NGS OneSeq kit (Agilent) data. Briefly, the cnvkit algorithm [26] was used with bam alignment files as the input. Filtering by a *p*-value of 0.001 was applied, and the copy number events were adjusted to the tumor burden of every sample; this tuning was applied during cns file creation. The genome’s LOH regions were established by comparing the heterozygote regions of a panel of five controls. Lastly, a post-analytical filter removing alterations shorter than 1 Mb, which were assigned as probable technical artifacts, was applied previously to the data analysis.

The studied parameters were the number of copy number variation (CNV) events, the average length per event, the length of the genome altered by these events, percentage of the genome altered, the same four parameters removing borderline events with a biallelic frequency (BAF) between 0.3 and 0.7, the number of gains, the length of the genome affected by gains, the percentage of genomes affected by gains, the three gain events removing a BAF between 0.3 and 0.7, the total number of losses, the length of genomes affected by copy number losses, the length of the genome suffering LOH events [27], the percentage of genomes altered by LOH, the number of events, and the length of genomes and the percentage of genomes altered with LOH spanning more than both 15 and 10 Mb. Continuous variables were categorized according to their median and quartiles.

The parameter settings and codes used for GI determination with cnvkit software and script to extract analytical features are available at https://github.com/afernandezse/Pola_Phase2_GI_traslational (accessed on 22 January 2022).

#### 2.4.3. Multiplex Ligation-Dependent Probe Amplification (MLPA) Analysis

To validate the in silico assessment of CN amplification and losses at the gene level in *CCNE1*, *PTEN*, and *EMSY* (previously described in the OC population), an MLPA analysis was performed. SALSA^®^ MLPA Probemix p225-E1 and P078-D2 Breast tumor assays were used, and the protocol was performed following the manufacturer’s instructions (MRC Holland, Amsterdam, The Netherlands). Amplified products were separated using an ABI3130XL Genetic Analyzer (Applied Biosystem, Foster City, CA, USA) and interpreted with GeneMapper Software v4.0 (Applied Biosystems, Foster City, CA, USA). Quantification of the fragment analysis results was performed using the Coffalyser software as described by the manufacturer (MRC Holland). Different normal control samples from healthy FFPE tissue were used to normalize the allele dosage.

The subrogates of the deficiency of the homologous recombination repair (HRR) pathway were HRD status, defined as single nucleotide variants (SNVs) and indels in HRR-genes, and different pre-established GI parameters (See the Appendix A). To assess their predictive power, these parameters faced response-based rates. ORR was the result of grouping CR and PR versus SD and PD, whereas Clinical Benefit Rate (CBR) grouped CR, PR, and SD of more than 6 months versus SD of fewer than 6 months and PD. Their associations with LTR were also explored.

### 2.5. Statistical Analysis

According to the Fleming method for phase 2 trials, 73 patients had to have more than a 90% chance of their ORRs being significantly different (0.05), considered the minimum (historical control 24%) and optimal ORRs for the proposed experimental schedule (estimated at 40%). For both safety and efficacy analyses, all patients who received at least one dose of the study treatment were included. The time-to-event analysis (PFS, OS, and duration of response) was analyzed using the Kaplan–Meier method. The Clopper–Pearson method was used to present the number and percentage of patients achieving a response with a two-sided 95% confidence interval (CI).

A statistical analysis was performed to define the correlations between clinicopathological and molecular parameters for time-to-event variables (i.e., PFS and OS). Differences between a Kaplan–Meier curves tests were determined using a log-rank test. R version 4.03 was used for statistics. A side effect was estimated using Cohen’s D test.

Further statistical analyses of all endpoints were performed following the Statistical Analysis Plan.

## 3. Results

### 3.1. Efficacy

A total of 84 patients were screened, and 73 patients received at least one dose of the study drugs (Appendix A). When the study database was locked (January 2019), all patients had discontinued treatment. The median treatment duration was 15 weeks (the minimum duration was 7 weeks, and the maximum was 25 weeks). The principal reason for treatment discontinuation was radiological progression disease in 59 patients (73%). Other reasons included patient decision (four patients), adverse event (two patients), death (one patient), protocol violation such as an overdose or skipped dose of olaparib (two patients), clinical progression (two patients), or clinical deterioration (three patients).

The clinical characteristics of the patients are summarized in Table 1. Most patients had HGS or high-grade endometrioid ovarian, fallopian tube, or primary peritoneum cancer (*n* = 46 patients, 63%), with endometrial cancer being the second most common tumor type (*n* = 26 patients, 35.6%). There was only one patient with triple-negative breast cancer. Twenty-seven (37%) patients presented visceral metastases at the time of the inclusion, with the most common site being lymph nodes in 16 patients (59.3%), followed by the lungs in 10 patients (37%). The median time from the diagnosis to the inclusion in the trial was 43.4 months (range: 6.3–171.8 months), and most patients (*n* = 45, 61.6%) had received three or more prior lines of therapy.

ORR evaluated per RECIST in the intention-to-treat population (73 patients) was 9.6%: one (1.4%) and six (8.2%) patients achieved CR and PR, respectively (Appendix A). However, the disease control rate (DCR = CR + PR + SD) was 72.6%. The best percentage of change from baseline in target lesions is shown in Figure 1. Five (6.8%) patients were unevaluable.

In the subgroup analysis, patients with endometrial cancer had higher ORR than patients with ovarian cancer (15.4% vs. 6.6%, respectively; *p* = 0.057). The number of previous lines of therapy influenced ORR; in patients with less than three previous lines (*n* = 28), the ORR was 21.4%, whilst in patients with three or more previous lines (*n* = 45), the ORR was 2.2% (*p* = 0.02). The sole patient with CR had been diagnosed with endometrial cancer and had received less than three lines of previous therapies.

The median PFS was 4.54 months (95% CI 3.0, 5.2) (Appendix A). No significant statistical differences were found in terms of PFS according to the primary site of the tumor. The median PFS was 4.5 (95% CI 3.0, 5.1) months for ovarian cancer and 4.8 (95% CI 1.9, 6.8) months for endometrial cancer (Appendix A). For the whole population, the PFS rate at 6 months was 28.56% (95% CI 18.34, 39.62) (Appendix A).

An exploratory analysis was performed to characterize the subset of patients deriving long-term benefit from the combination of lurbinectedin and olaparib. Long-term responders (LTRs) were defined as patients whose PFS was equal to or greater than the double estimated median PFS (4.54 months). In total, 12 (16.4%) of the 73 patients were considered LTR, with a median PFS of 13.3 months (Appendix A). The median OS for the entire population was 15.19 (95% CI 12.13, 17.69) (Appendix A). No differences were found in OS according to the tumor type (Appendix A).

### 3.2. Safety and Tolerability

All 73 patients had at least one treatment-emergent adverse event (TEAE) (Table 2). Overall, treatment with lurbinectedin and olaparib was well-tolerated, with most TEAEs being grade 1 or 2. A total of 26 patients (35.5%) experienced grade 1–2 TEAEs, most commonly asthenia, nausea, vomiting, constipation, diarrhea, abdominal pain, dysgeusia, and anemia. The most common grade > 3 TEAE was hematological toxicity, predominantly neutropenia, which was reported in 28 patients (38.3%). Grade 3–4 anemia and thrombocytopenia were observed in 6 (8.2%) and 3 (4.1%) patients, respectively. The most common grade 3–4 non-hematologic toxicity was asthenia, reported in 6 patients (8.2%). Serious TEAEs were observed in 22 patients (30.1%), which were related to study drugs in 3 of the patients: 1 patient had grade 3 diarrhea, 1 patient had grade 3 constipation, and 1 patient had grade 3 cardiac disorders. No deaths were related to adverse events.

Over the course of the treatment, 42 patients (57.5%) required a dose reduction in at least one drug due to adverse events. Six patients (8.2%) discontinued treatment due to toxicity as the main reason.

### 3.3. Translational Studies

#### 3.3.1. Distribution of Genetic Alterations and Clinical Impact

Genetic studies were performed on a total of 57 samples that passed the quality and quantity requirements, corresponding to 19 (33.3%) endometrial cancer and 38 (66.7%) ovarian cancer patients. Among all of the mutated genes, considering both cancer types, *TP53* and *PTEN* presented the highest mutational ratios, with 34/57 (59.6%) and 9/57 (15.8%), respectively, excluding CNVs. *TP53* alterations were mainly present in ovarian cancer (70.6%), specifically in HGS histology, while *PTEN* was preferentially altered in endometrial cancer (88.9%). Regarding the HRR pathway, a total of eight genes presented alterations, including *BRCA1* (3, 5.3%), *BRCA2* (1, 1.8%), *ATM* (2, 3.5%), *RAD5L* (1, 1.8%), *ATR* (1, 1.8%), *NBN* (1, 1.8%), *SLX* (1, 1.8%), and *WRN* (1, 1.8%). Overall, HRR gene alterations were reported in 10/57 (17.5%) cases homogeneously distributed between endometrial cancer and ovarian cancer, and they were used in the following analysis as an HRD status subrogates. Additionally, mutations in the Fanconi Anemia genes, *FANCM* (1, 1.8%) and *FANCA* (1, 1.8%), were also found. Finally, alterations in the *MMR* genes were described in two endometrial cancer cases (Figure 2).

We studied the possible relationship between HRD status and response to treatment. No correlations were found between study treatment ORR or CBR, and HRR mutations. The different GI parameters (Appendix A and Methods) were compared with the mutation-based stratification. In the whole population, HRD status was associated with losses (*p* = 0.0038) and the percentage of the genomes affected by losses (*p* = 0.034) (Figure 3A,B). Considering that GI caused by HRR gene mutations has been principally described in the ovarian cancer population, we studied GI patterns according to cancer type (Appendix A). The ovarian cancer cohort (*n* = 38) showed a significant correlation between HRD status and the total number of events (*p* = 0.0053), loss events (*p* = 0.0012), and percentage of the genome affected by losses (*p* = 0.012). Loss of heterozygosity (LOH) did not correlate with treatment response (*p* = 0.091) (Figure 3C–F). On the other hand, the endometrial cancer cohort (*n* = 19) did not show any significant results.

#### 3.3.2. Characterization of Copy Number Patterns across the Clinical Trial Population: Clinical Impact of Genomic Instability-Based Classification

Finally, the GI parameters were evaluated as a predictive biomarker for the combination of olaparib and lurbinectedin. First, in terms of the response and duration of response, LTRs were assessed. When evaluating the ovarian cancer population specifically (*n* = 27), we observed a trend towards an association between LTRs and total LOH events, which did not reach statistical significance with the current sample size (*p* = 0.055) (Figure 4B). Second, the relationship between GI and ORR was also evaluated. The total number of LOH events was not associated with ORR (*p* = 0.074) (Appendix A). We observed a significant correlation between ORR and the percentage of genome altered by losses (*p* = 0.021), although only two cases qualified as responders with HGS histology (Figure 4A). In the endometrial cancer population, the percentage of the total genome that was altered was not associated with ORR (*p* = 0.07) (Appendix A). Finally, the classification of responses as CBR was studied, but did not yield significant associations, for example, with the total number of events (*p* = 0.063) and gains (*p* = 0.088) (Appendix A). In the HGS population (*n* = 38), a higher number of events was significantly associated with longer PFS (*p* = 0.041) (Figure 4C). Although the GI parameters were correlated with the PARPi response in the non-parametric tests, only few parameters showed significance in the univariate survival analysis, and multivariate analysis was not significant. However, the results showed a correlation between higher GI and outcome, which raises the possibility of developing this parameter as a predictive marker.

## 4. Discussion

The combination of an inhibitor of DNA damage repair, such as olaparib, with a DNA damaging agent, such as lurbinectedin, is an exciting approach to maximizing the effect of DNA damage. In preclinical models, the combination of olaparib and lurbinectedin has shown a synergistic effect with relevant antitumor activity [21]. However, overlapping toxicities make the combination difficult. Hematological toxicity is the major concern of this combination, since only lurbinectedin as therapy showed grade 3–4 neutropenia up to 85% when administered at a flat dose [19], though this was lower (57%) when the dose was adjusted to body surface area [23]. On the other hand, treatment with chemotherapy and continuous doses of olaparib is usually not feasible due to the high rate of hematologically adverse events. An intermittent schedule of olaparib is better tolerated than a continuous one when combined with chemotherapy [10,11]. In our study, the treatment with lurbinectedin and olaparib was tolerable. Compared with phase 1, no new threats to safety in the expanded phase 2 study were observed. The most common adverse events were hematological (38% of patients had neutropenia grade ≥ 3), and among the nonhematological events, the most common was asthenia, in 8.2%. Although dose modification of at least one drug due to adverse events was common (57.5%), only six patients (8.2%) discontinued treatment due to toxicity.

In the POLA phase 1 dose-escalation trial, we demonstrated that the combination of lurbinectedin adjusted to body surface area and a short course of olaparib had a safe and tolerable profile with an encouraging DCR (stable disease 60%) in a heavily pretreated population [24]. In this phase 2 study, we assessed the efficacy of the RP2D of lurbinectedin (1.5 mg/m^2^ on day 1) with a short course of olaparib (250 mg twice a day on days 1–5) administered every three weeks. To our knowledge, this study is the first phase 2 trial that tests this combination in gynecological malignancies. We showed that this combination provides an ORR of 9.6%, below the pre-specified boundary of efficacy (40%) and even in historical controls (24%). However, DCR was 72.6%, and 12 (16.4%) of the 73 patients treated were considered LTRs, with a median PFS of 13.3 months. In our study, there is particular difficulty in estimating the efficacy across different tumor types and patient characteristics: 61.6% of the patients were heavily pretreated (three or more lines of treatment) since no limit on previous lines of therapy was established. This population had a worse prognosis than the populations included in the recent large phase 3 studies: ovarian cancer was limited to three previous lines, and endometrial cancer was limited to two previous line. Our study subgroup analyses were performed according to histology, and the number of previous lines of therapy showed that patients with three or more previous lines of therapy had a low probability of response (2.2%). In our study, ovarian cancer patients had an ORR of 6.6%, irrespective of histology (HGS or endometroid) and HR status. Historically, the four drugs (pegylated liposomal doxorubicin, paclitaxel, gemcitabine, and topotecan) most often used as single agents in platinum-resistant ovarian cancer had similar response rates (ranging from 10% to 15%) [28]. In select patients, the addition of bevacizumab to chemotherapy increases their response rates [29], but in most of these trials, the patients had received only one or two lines of previous chemotherapy. In a phase 2 study, lurbinectedin showed significant improvement in ORR compared with topotecan [19]. Regarding PARPi, olaparib monotherapy was approved by the FDA for patients treated with three or more lines based on the results (ORR of 34% and median PFS of 7.9 months) of a series of patients with *BRCA* mutations and platinum-resistant disease [30]. In the ROLANDO trial [31], olaparib combined with pegylated liposomal doxorubicin was assessed in platinum-resistant ovarian cancer and showed an ORR of 29%. However, the number of prior therapy lines was limited to a maximum of four, with at least one previous platinum-sensitive relapse, and *BRCA* mutations were present in 16% of patients compared with 7% of patients in the present study. On the other hand, patients with endometrial carcinoma had an ORR of 15.4%, which is higher than that reported with other PARPis in monotherapy, such as niraparib (4%) [32]. For endometrial cancer, the most active chemotherapeutic agents identified have been doxorubicin and cisplatin, and both alone or in combination with other agents have been tested in phase III trials, with ORRs ranging from 14.7% to 42% [29,33,34,35,36]. However, lurbinectedin plus doxorubicin has only been tested in a small phase I trial with an ORR of 42% [37]. However, the median chemotherapy lines for advanced disease in this population was 1 (range 0–2).

Historically, cancer treatments have been investigated without studying biomarkers of response or fully understanding the mechanisms underlying resistance to the treatment. However, recent trials have evidenced the role specific biomarkers have played in the development of new treatments. The POLA translational study was designed to describe the correlation between different GI parameters and the benefit of the treatment, establishing GI as a predictive biomarker in this clinical scenario. In total, 57 cases were evaluated at the gene and genomic levels, defining those that presented HRD based on the mutational status of HRR genes (10 patients, 17.5%). *BRCA1/2* tumor mutations were present in 10% of cases, which is below the 20% of germline and somatic cases reported in HGS ovarian carcinoma [30]. Due to the nature of the population, with patients who have been previously treated with PARPis and, therefore, potentially mutated and responsive patients being excluded, the incidence of cases with *BRCA* mutations suffered an evident decrease. Regarding the correlation between HRD classification based on mutational status and response, no significant association was found. However, we found a significant correlation between response and different GI parameters, such as loss events, mainly present in the ovarian cancer cohort and HRD population. The lack of predictive power of HRR gene mutations could be explained by differences regarding the characteristics of the population, given that the population is composed principally of patients with HGS ovarian cancer who have been heavily pretreated and a lower frequency of *BRCA* mutations compared with other series. Recent reports have evidenced that mainly cases harboring *BRCA* mutations and, marginally, other HRR genes, such as *RAD51C* conferred sensitivity to PARPi [31,32]. Clinical and methodological issues might also have an impact on the results. For instance, the fact that the genetic and genomic analysis was performed on the primary tumor and not at the moment of relapse, previously to study entry, could affect the concordance between HRD status and treatment response. The mutational/LOH patterns are not reverted when a tumor recovers HR function, so they may not be accurate in predicting PARPi sensitivity in patients who have previously received and progressed on DNA damaging chemotherapy, such as platinum. In addition, the variant selection, which includes in silico prediction of pathogenesis, could have an impact on sensitivity prediction, since some of those mutations may not have a real loss of function and, hence, may not present the HRD phenotype.

The lack of a gold standard for the definition and assessment of GI has motivated a wide number of studies to find an accurate approach [33]. However, only two of them have been commercially approved: Myriad MyChoice^®^ and the one from Foundation Medicine^®^. Even if both approaches have been extensively validated [6,31], developing an in-house tool adapted to our requirements and being able to establish the GI based on different parameters were advantages. Additionally, for the mutational analysis, we assessed the whole-genome CNV phenotype and adjusted an in-house pipeline to interrogate and define the GI patterns with regard to the combined treatment response. Hence, we aimed to achieve the most suitable classifier, overcoming the possible caveats of available methodologies. Our approach showed a correlation between different GI parameters and better response to the studied combination. The results concerning the ovarian cancer population were particularly interesting, where a higher percentage of losses (*p* = 0.021) appeared to be correlated with ORR. At the same time, without reaching statistical significance, a trend was observed between the number of LOH events (*p* = 0.055) with LTRs. However, all these results should be carefully considered because of the limited sample size. In addition, the total number of events was also significant in the log-rank test (Figure 4). As several GI parameters are associated with better responses, our next steps will be focused on obtaining a model combining these pre-defined parameters, using response as the endpoint. Then, the predictive role of this GI model will need to be validated in a prospective trial specifically addressing this endpoint.

## 5. Conclusions

In conclusion, the POLA study provides evidence that the administration of 1.5 mg/m^2^ of lurbinectedin on day 1 and 250 mg of olaparib twice a day on days 1–5 every 21 days is feasible with a DCR of 72.6% and tolerable safety profile in patients who have been heavily pretreated for gynecological cancer. Based on these results, the combination would be suitable for further research and offers a potential alternative for patients with relapsed ovarian and endometrial cancer irrespective of *BRCA* mutation status. This translational study showed a correlation between different GI parameters and a better response; however, its predictive impact should still be investigated in a larger randomized study.

## Figures and Tables

**Figure 1 cancers-14-00915-f001:**
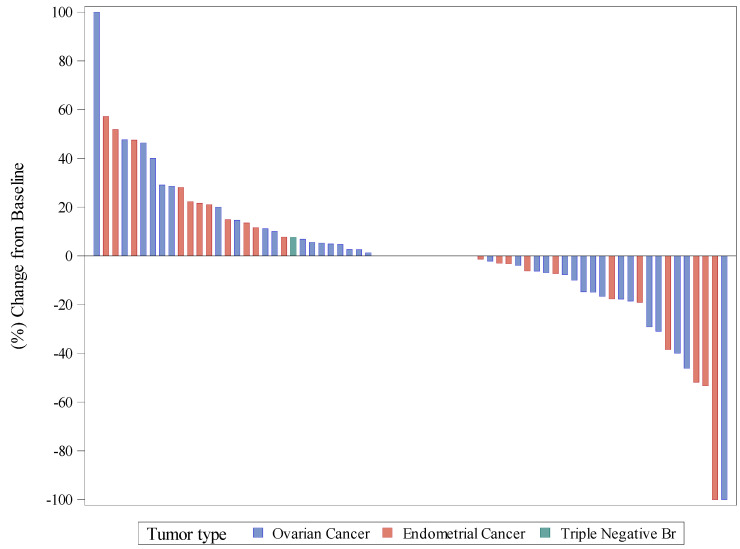
Waterfall plot of best response, as a percentage of change in target lesions.

**Figure 2 cancers-14-00915-f002:**
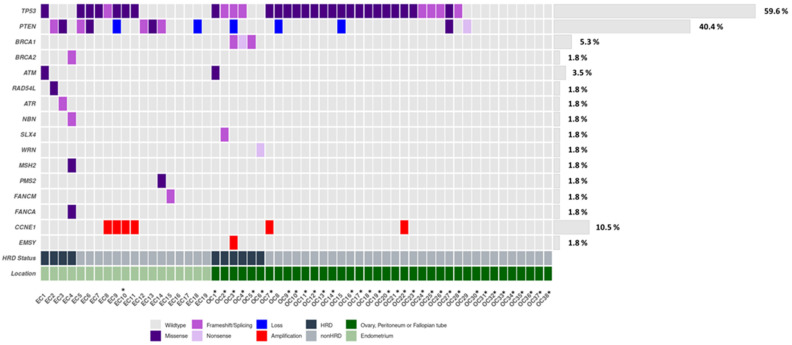
Oncoplot of genetic and genomic alterations across the 35 genes of the custom panel, related to HRR and other DNA repair pathways across the EC (*n* = 19) and OC (*n* = 38) cohorts. The Oncoplot shows SNVs and CN at the gene level in *CCNE1*, *PTEN*, and *EMSY*. * Serous histology.

**Figure 3 cancers-14-00915-f003:**
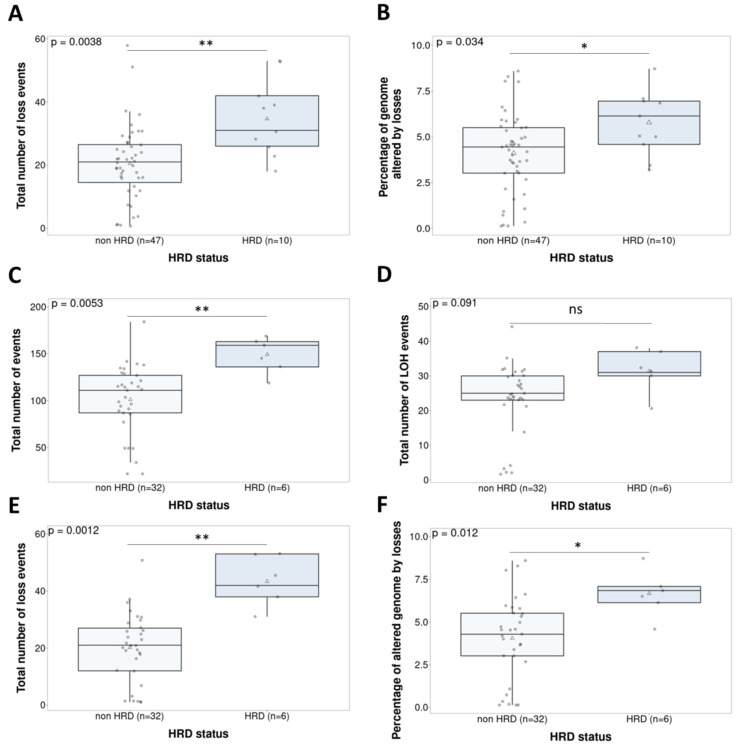
Non-parametric tests (Wilcoxon signed rank test) comparing the genomic instability parameters and HRD status in the whole population (*n* = 57)—loss events (Cohen’s d = 1.2) (**A**) and percent of altered genome by losses (d = 0.8) (**B**)—and in the ovarian cancer population (*n* = 38)—total number of events (d = 1.4) (**C**), LOH events (d = 0.81) (**D**), loss events (d = 2) (**E**), and percent of altered genome by losses (d = 1.17) (**F**). ns. Not significant; * *p* < 0.05; ** *p* < 0.01.

**Figure 4 cancers-14-00915-f004:**
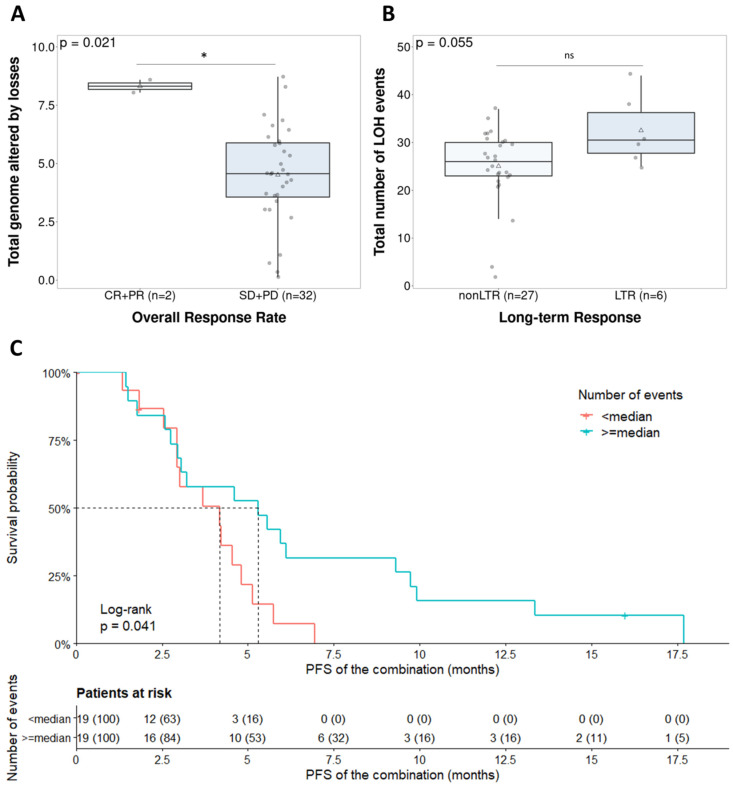
Clinical implication of the genomic instability parameters in the ovarian cancer population (*n* = 38). (**A**) Total number of events regarding overall response rate (d = 0.69); (**B**) total number of LOHs in long-term responders (d = 0.31); (**C**) survival analysis stratification due to the total number of events. Ns: Not significant; * *p* < 0.05.

**Table 1 cancers-14-00915-t001:** Patient characteristics.

Characteristic	Lurbinectedin plus Olaparib, *n* = 73 Patients
Age, median (range), years	65 (22–80)
Gender	
Females, *n* (%)	73 (100)
ECOG PS, *n* (%)	
0	40 (54.8)
1	33 (45.2)
Primary tumor type, *n* (%)	
Ovarian carcinoma	46 (63)
High-grade serous	44 (60.3)
High-grade endometroid	2 (2.7)
Endometrial carcinoma	23 (31.5)
Endometrial carcinosarcoma	3 (4.1)
Triple negative breast cancer	1 (1.4)
Metastasis at baseline, *n* (%)	27 (37)
Lung	10 (37)
Liver	5 (18.5)
Lymph nodes	16 (59.3)
Bone	1 (3.7)
Others	12 (44.4)
Number of previous treatment regimens	
<3 treatments, *n* (%)	28 (38.4)
≥3 treatments, *n* (%)	45 (61.6)

**Table 2 cancers-14-00915-t002:** Treatment-emergent adverse event (≥5%) by maximum grade per patient.

TEAE	Grade 1		Grade 2		Grade 3		Grade 4		Total	
	*n*	(%)	*n*	(%)	*n*	(%)	*n*	(%)	*n*	(%)
Anemia	8	(10.9)	7	(9.5)	6	(8.2)	0	(0)	21	(27.3)
Leukopenia	3	(4.1)	5	(6.8)	3	(4.1)	0	(0)	11	(15)
Neutropenia	0	(0)	10	(13.6)	19	(26)	9	(12.3)	38	(52)
Trombocytopenia	4	(5.4)	2	(2.7)	2	(2.7)	1	(1.3)	9	(12.3)
Abdominal pain	14	(19.0)	10	(13.6)	2	(2.7)	0	(0)	26	(35.3)
Constipation	18	(24.6)	6	(8.2)	0	(0)	0	(0)	24	(32.8)
Diarrhea	15	(20.5)	4	(5.4)	1	(1.3)	0	(0)	20	(27.3)
Dyspepsia	3	(4.1)	2	(2.7)	0	(0)	0	(0)	5	(6.8)
Nausea	30	(41)	11	(15)	0	(0)	0	(0)	41	(56.1)
Vomiting	12	(16.4)	7	(9.5)	1	(1.3)	0	(0)	20	(27.3)
Asthenia	15	(20.5)	29	(39.7)	6	(8.2)	0	(0.0)	50	(68.4)
Fatigue	2	(2.7)	3	(4.1)	0	(0)	0	(0)	5	(6.8)
Mucosal inflammation	3	(4.1)	1	(1.3)	0	(0)	0	(0)	4	(5.4)
Pyrexia	9	(12.3)	1	(1.3)	0	(0)	0	(0)	10	(13.6)
Bronchitis	3	(4.1)	2	(2.7)	0	(0)	0	(0)	5	(6.8)
Urinary tract infection	1	(1.3)	0	(0)	0	(0)	0	(0)	1	(1.3)
ALT/GPT increased	4	(5.4)	1	(1.3)	1	(1.3)	0	(0)	6	(8.1)
AST/GOT increased	5	(6.8)	2	(2.7)	0	(0)	1	(1.3)	8	(10.9)
GGT increased	0	(0)	1	(1.3)	0	(0)	3	(4.1)	4	(5.4)
Decreased appetite	8	(10.9)	4	(5.4)	0	(0)	0	(0)	12	(16.4)
Hypoalbuminaemia	2	(2.7)	4	(5.4)	0	(0)	0	(0)	6	(8.2)
Hypogamnesaemia	7	(9.5)	0	(0)	0	(0)	0	(0)	7	(9.5)
Artralgia	4	(5.4)	1	(1.3)	0	(0)	0	(0)	5	(6.8)
Back pain	7	(9.5)	3	(4.1)	0	(0)	0	(0)	10	(13.6)
Dizziness	3	(4.1)	0	(0)	0	(0)	0	(0)	3	(4.1)
Dysgeusia	16	(21.9)	1	(1.3)	0	(0)	0	(0)	17	(23.2)
Cough	4	(5.4)	0	(0)	0	(0)	0	(0)	4	(5.4)
Dyspnea	6	(8.2)	4	(5.4)	1	(1.3)	0	(0)	11	(15)
Pulmonary embolism	1	(1.3)	1	(1.3)	4	(4.5)	0	(0)	6	(8.2)
Lymphedema	3	(4.1)	1	(1.3)	0	(0)	0	(0)	4	(5.4)

## Data Availability

Data is contained within the article or Appendix A.

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
