# Peer review of "Phase 2 Trial (POLA Study) of Lurbinectedin plus Olaparib in Patients with Advanced Solid Tumors: Results of Efficacy, Tolerability, and the Translational Study"

_cancers, 2022, doi:10.3390/cancers14040915_

Round 1
Reviewer 1 Report
This is an interesting study using a novel treatment approach. Though the primary endpoint was not reached, it is still important to publish "negative" trials. I understand the motivation to spin this trial more positively, but I think it needs to be clear that DCR was not a pre-specified endpoint and that the duration of disease control rate is very short and may not be clinically relevant.
I also don't completely understand the rationale for the short course olaparib. I understand that it is more tolerable, but is there any efficacy data? I think presenting that would help justify the rationale for this study.
Additional comments:
Introduction: I think it would be helpful if the authors described the mechanism of lurbinectedin and a short phrase re: its use in platinum resistant lung cancer. Are there pre-clinical data that explain the rationale for this drug combination? A short explanation might be helpful. Since PARP inhibitors are not typically expected to be effective in platinum resistant gynecologic cancers, I think it’s important to explain the rationale. I see it is in the discussion, but maybe just a short explanation would fit in the introduction to give readers the context. Most gyn oncs will not be familiar with lurbinectedin.
Line 205: Does “distant” metastases here indicate extra-abdominal metastases? I would clarify this because I’m assuming all of these patients have metastatic disease, so it’s confusing to see that only 37% have metastatic disease.
Line 215: Would change “no evaluable” to “unevaluable”
Line 277: I don’t understand what this sentence means. I think it is describing that HRD was associated with GI?
Line 295: These p-values are non-significant. Can you add clarification to specify this?
Line 301-303: What is meant by “events?” Can you explain this in more detail in the manuscript?
Line 345-350: I think this paragraph needs to be organized differently. It might help if you listed the historic response rate to chemotherapy in ovarian cancer first, then the response rate to lurbinectedin in phase I or other studies in ovarian cancer, then the response rate for PARPi in platinum resistant ovarian cancer (or more specifically HRD, BRCA, or GI ovarian cancer). Next, you could list the historic response rate to chemotherapy in endometrial cancer, then any published response rates to lurbinectedin, followed by any published response rates to PARPi monotherapy. As it is, it’s very confusing trying to compare these numbers. Additionally, it appears the response rate in this trial was actually lower than lurbinectedin alone, so it’s hard to justify further study. It brings up the question as to whether this short course PARPi has any clinical activity.
Author Response
Attached you can find the answers to reviewer 1

Reviewer 2 Report
Poveda et al report the outcomes and translational studies from single-arm POLA Phase 2 clinical trial which combines Olaparib with lurbinectedin in solid tumors. Although the study was negative, the results are of interest, and important to the gynecologic oncology field. The manuscript is written clearly and concisely, and the conclusions are mostly supported by the data. The authors nicely discuss the limitations of the study. There are a few concerns that need to be addressed. Major comments- The justifications to combine Olaparib with lurbinectedin are missing from the introduction. Also mechanisms and background of lurbinectedin should be added to the introduction.
- As the genomic instability metrics remain the most relevant translational finding, the authors should i) explain in more detail in the main manuscript how the GI was assessed. For instance which features are the 14 parameters established a priori? Are they in line with the established HRD scores as in Telli et al 2016. ii) provide the algorithm/code available for review purposes, to allow for more thorough assessment of the GI.
- Statistics: are the p-values corrected for multiple hypothesis testing? Effect sizes should be calculated for the box plots in Figure 3, Supplementary figures S4-6. Are the different GI values intercorrelated? Would a multivariate analysis reveal which of them (if any) is actually significant? In small studies such as this, the “borderline significances" should be considered carefully.
Author Response
Attached you can find the answers to reviewer 2

Reviewer 3 Report
The manuscript describes the outcomes of a phase 2 clinical trial of lurbinectedin plus olaparib in patients with advanced solid tumors that includes mutational analysis of a large part of the patient population studied.
Overall the manuscript is well written and provides valuable insights into the application of olaparib treatment in heavily pretreated ovarian and breast cancer patients. However, the authors attach too many important conclusions to data that is non-significant, even when considering a cutoff of P < 0.05. The manuscript should be extensively rewritten to set a focus on the actually significant findings and to not provide misleading conclusions. Negative results can still provide valuable information about treatment efficacy when stated as such.
Major comments:
The authors need to distinguish better between significant (P<0.05) and non-significant results.
e.g. line 281-283 ...significant correlation...loss of heterozygosity (LOH) (p=0.091)
e.g. line 293-295 LTRs showed a direct association with GI when evaluated in the ovarian cancer population, particularly with total LOH events (p=0.055)
Indicate significance in graphs with */**/*** rather than just p value to again make clear difference between significant and non-significant.
Conclusions cannot be based on non-significant results.
e.g. line 47-48 Globally, the total number 47 of LOH events seems associated with ORR (p=0.074)
Minor comments:
Please introduce lurbinectedin and its use and efficacy for ovarian/breast cancer treatment.
Addition of a list of abbreviations is warranted.
Author Response
Attached you can find the answers to reviewer 3

Round 2
Reviewer 2 Report
The authors have addressed most of my comments. Two things need to be addressed to warrant publication in Cancers. The following need to be addressed prior to acceptance: 1) The individual data points have not been added to the box plots in figures 3, and S4-6. 2) Figure 1 needs to be coloured by the corresponding tumor types. The authors explain they will be able to add this in mid February, and I do think that it would be critical for the acceptance of the manuscript.Author Response
Attached you can find the answer to your comments and suggestions.

Reviewer 3 Report
I would like to thank the authors for revising the manuscript. Especially the introduction and discussion parts are much improved and grant a better understanding of the study. Upon adjustment of the below statements, the manuscript will be suitable for publication in Cancers.
Comments on the revised version of the manuscript ‘Phase 2 trial (POLA study) of lurbinectedin plus olaparib in patients with advanced solid tumors: results of efficacy, tolerability and the translational study’:
Again, please be entirely clear about the results’ significance. If the p value is not significant (P < 0.05 is a strict cutoff), a correlation is by definition not present, and this should be stated so. Consideration of a p value of more than 0.05 is not scientifically sound especially with the small sample sizes used in some of the comparisons. Please rephrase the according sentences as stated below (after checking my suggestions for correctness) and make mention the issue of small sample size in the discussion.
For additional readability, it would also be advantageous to add all individual data points to the box-whisker plots (Figures 3, 4, S4, S5, S6) in order to reliably reflect that the sample sizes are very different between the various graphs. Please also state clearly which exact populations are included in each graph in the Figure legends.
Results section:
The ovarian cancer cohort showed a significant correlation between HRD status and the total number of events (p=0.0053), loss events (p=0.0012), and percentage of the genome affected by losses (p=0.012). However, correlation with loss of heterozygosity (LOH) was not statistically significant (p=0.091), (Figure 3C, D, E, and F). On the other hand, the endometrial cancer cohort did not show any significant results.
> The ovarian cancer cohort (n = 38) showed a significant correlation between HRD status and the total number of events (p=0.0053), loss events (p=0.0012), and percentage of the genome affected by losses (p=0.012). Loss of heterozygosity (LOH) did not correlate with treatment response (p = 0.091). On the other hand, the endometrial cancer cohort (n = 19) did not show any significant results.
LTRs showed a direct association with GI when evaluated in the ovarian cancer population, particularly with total LOH events although this did not reach statistical significance (p=0.055) (Figure 4B).
> When evaluating the ovarian cancer population specifically (n = 27), we observed a trend towards association of LTRs with total LOH events, but this did not reach statistical significance with the current sample size (p=0.055) (Figure 4B).
Globally, the total number of LOH events was associated with ORR (p=0.074) (Supplementary Figure S5A). The percentage of genome altered by losses (p=0.021) (Figure 4A) and the total genome altered (p=0.07) (Supplementary Figure S5B) showed correlation with ORR in HGS histology and endometrial cancer, respectively.
> The total number of LOH events was not associated with ORR (p=0.074) (Supplementary Figure S5A). We observed a significant correlation of ORR with the percentage of total genome altered by losses (p=0.021) although it should be noted that only 2 cases qualified as responders with HGS histology (Figure 4A). In the endometrial cancer population, the percentage of total genome altered was not associated with ORR (p = 0.07)
Finally, classification of response as CBR was studied. In this case, only the HGS population sub-analysis presented some statistical trends, among them, the total number of events (p=0.063) and gains (p=0.088) (Supplementary Figure S6)]. Additionally, in the HGS population, survival analysis regarding the total number of events also appears significant (p=0.041) (Figure 4C).
> Finally, classification of response as CBR was studied but did not yield significant associations for example with the total number of events (p=0.063) and gains (p=0.088) (Supplementary Figure S6). In the HGS population (n = 38), a higher number of total events was significantly associated with longer PFS (p=0.041) (Figure 4C).
Although GI parameters correlated with the PARPi response in non-parametric tests, the univariate survival analysis presented limited significance while multivariate analysis was not significant.
> please add a table with the results of this analysis and explain what you mean by “limited significance” (please adhere to a strict cutoff of p < 0.05 for stating significance!)
Discussion section:
> Please integrate the newly added part with the findings of the current study. For example, in lines 369-375 the previous findings on hematological toxicity in clinical trials using lurbinectedin are discussed. This part would be well integrated with the hematological side effects seen in this study as stated in lines 417-421 to make a direct comparison easier for the reader.
For instance, the fact that the genetic and genomic analysis was performed on the primary tumor and not at the moment of relapse, previously to initiate treatment in combination, could affect the concordance between HRD status and treatment response.
> do you mean ‘prior to combination treatment initiation’? Meaning of sentence part in bold is unclear.
Our approach showed a correlation between different GI parameters and better response to the studied combination
> please state the specific parameters instead of ‘different GI parameters’ and state only truly significant (p < 0.05!) results. The marginally significant result regarding the higher number of LOH events (p=0.055) may be left here if cautioning that a larger sample size is necessary to reveal whether a true correlation is present.
Author Response
Attached you can find the answer to your comments and suggestions.
